# Liquid Biopsies as Non-Invasive Tools for Mutation Profiling in Multiple Myeloma: Application Potential, Challenges, and Opportunities

**DOI:** 10.3390/ijms25105208

**Published:** 2024-05-10

**Authors:** Robbe Heestermans, Rik Schots, Ann De Becker, Ivan Van Riet

**Affiliations:** 1Department of Clinical Biology, Vrije Universiteit Brussel (VUB), Universitair Ziekenhuis Brussel (UZ Brussel), Laarbeeklaan 101, 1090 Brussels, Belgium; 2Department of Hematology, Vrije Universiteit Brussel (VUB), Universitair Ziekenhuis Brussel (UZ Brussel), Laarbeeklaan 101, 1090 Brussels, Belgium; 3Translational Oncology Research Center (Team Hematology and Immunology), Vrije Universiteit Brussel (VUB), Laarbeeklaan 103, 1090 Brussels, Belgium

**Keywords:** liquid biopsy, multiple myeloma, mutation profiling, cell-free DNA, circulating tumor cells, extracellular vesicles, personalized medicine

## Abstract

Over the last decades, the survival of multiple myeloma (MM) patients has considerably improved. However, despite the availability of new treatments, most patients still relapse and become therapy-resistant at some point in the disease evolution. The mutation profile has an impact on MM patients’ outcome, while typically evolving over time. Because of the patchy bone marrow (BM) infiltration pattern, the analysis of a single bone marrow sample can lead to an underestimation of the known genetic heterogeneity in MM. As a result, interest is shifting towards blood-derived liquid biopsies, which allow for a more comprehensive and non-invasive genetic interrogation without the discomfort of repeated BM aspirations. In this review, we compare the application potential for mutation profiling in MM of circulating-tumor-cell-derived DNA, cell-free DNA and extracellular-vesicle-derived DNA, while also addressing the challenges associated with their use.

## 1. Introduction

Multiple myeloma (MM) is a hematological malignancy characterized by the proliferation and accumulation of monoclonal, neoplastic plasma cells in the bone marrow, generally associated with the presence of monoclonal proteins in blood and/or urine and end-organ damage [1,2]. The typical clinical features of MM include hypercalcemia, osteolytic bone lesions, anemia, and renal insufficiency, often referred to as CRAB lesions. The incidence of MM has increased over the past three decades, with over 176,000 incident cases in 2020 worldwide and is expected to further increase coinciding with the population ageing [3,4]. It is well known that the incidence of MM is higher among men and non-Hispanic black people, which has been attributed to a higher incidence of MGUS in these populations [5,6]. Mainly because of the introduction of autologous stem cell transplantation and novel therapeutic agents such as proteasome inhibitors and immunomodulatory drugs during the past decades, survival among MM patients has considerably improved. However, patients continue to relapse, and, in particular, those with high-risk disease characteristics including unfavorable cytogenetic abnormalities have inferior outcomes [7,8,9,10,11]. The increasing use of next-generation sequencing (NGS) technology over the last decade has shown that in many MM patients, a complex, dynamic tumor mutation profile occurs that affects prognosis and therapeutic response [12,13,14,15,16,17]. In addition, several studies have demonstrated a spatial genetic heterogeneity in the BM compartment of MM patients. This may potentially lead to an incomplete assessment of the genetic heterogeneity when using conventional single-site bone marrow aspirates [18,19,20]. As a result, there is an increasing interest in the use of “liquid biopsies”, consisting of blood samples from which circulating tumor markers such as circulating tumor cells (CTCs) and cell-free DNA (cfDNA) can be isolated to characterize the genetic tumor profile. Although this approach seems promising and creates new possibilities in an era where precision medicine increasingly becomes the standard of care, questions regarding feasibility and reliability also need to be addressed. In this review, we therefore aim to summarize the current knowledge about the use of liquid biopsies for mutation profiling in MM, discuss the challenges associated with their use, and look ahead towards new applications of liquid biopsies in MM.

## 2. Molecular Characterization in Multiple Myeloma and Impact on Prognosis and Therapy Response

The development of MM is preceded by an asymptomatic plasma cell proliferation, called a Monoclonal Gammopathy of Unknown Significance (MGUS), representing the earliest stage of this complex disease. Primary genetic events appear to take place in the germinal center of lymph nodes, facilitated by a phase of somatic hypermutation and the isotype switching of maturating B-cells. The initial cytogenetic alterations driving the development of MGUS consist of IgH translocations that implicate and deregulate oncogenes such as *CCND1*, *MMSET*, *MAF* (=non-hyperdiploid type), and trisomies of chromosomes 3, 5, 7, 9, 11, 15, 19, and 21 (=hyperdiploid type) (Figure 1) [21]. Importantly, according to the International Myeloma Working Group (IMWG), the t(4;14), t(14;16), t(14;20), del(17/17p), non-hyperdiploidy, and gain(1q) are cytogenetic abnormalities that are associated with inferior prognosis [22]. Although these key genetic changes are already detectable in patients with MGUS, somatic variants occur at lower frequencies and with a lower degree of complexity in the early stages of plasma cell dyscrasia and are typically acquired throughout progression towards more advanced disease stages [23,24,25]. At least one mutation in the MAPK pathway, containing the oncogenes *NRAS*, *KRAS*, and *BRAF*, occurs in ±50% of all MM patients [25,26]. Genes associated with the NF-kB (e.g., *TRAF3*, *LTB*, *CYLD*) and DNA repair pathways (e.g., *ATM*, *ATR* and *TP53*) are mutated in ±20% of MM patients and are also among the most commonly affected in MM [12,27,28,29]. Germline variants in *DIS3*, *KDM1A*, *ARID1A*, and *USP45* in families with multiple cases of MM/MGUS are associated with an increased risk of MM development [30,31,32]. However, their exact role in MM pathogenesis has not yet been elucidated.

Currently, mutation profiling using NGS is not routinely performed in the diagnostic work-up of MM, and genetic characterization is still mainly focused on the detection of the above-mentioned cytogenetic alterations. Instead, the use of NGS in clinical practice is directed towards the detection of Measurable Residual Disease (MRD) in MM via the detection of the tumor-specific rearrangements in the IgH, IgK, and/or IgL loci [33,34] It has been shown that obtaining MRD negativity is strongly associated with an improved survival outcome, with the optimal threshold for MRD sensitivity set at 10^−6^ [35]. However, with whole-genome sequencing becoming increasingly accessible in clinical practice, there is a growing interest in the characterization of mutations associated with therapy resistance and prognostic impact. Mutations in the *PSMB5* gene are implicated in resistance to proteasome inhibitors (PI, e.g., bortezomib), while mutations in *CRBN* and genes of the Cereblon pathway are associated with resistance to treatment with immunomodulatory agents (IMiDs) [36,37]. Only very recently, Perroud et al. showed that mutations in the MAPK pathway are associated with unfavorable outcomes in patients treated with PI/IMiD combinations [15]. Furthermore, it has been reported that mutations in *TP53*, *ATM*, *ATR*, and *CCND1* confer to an inferior prognosis, while *IRF4* mutations have been associated with a favorable prognostic impact [12]. These findings highlight the potential impact of mutation profiling to assess prognosis and the chances of therapeutic success more accurately and guide clinicians in a personalized therapy decision-making process. 

## 3. Liquid Biopsies: A Comprehensive and Non-Invasive Alternative to Bone Marrow Aspirates

MM is characterized by a patchy tumor infiltration pattern in the bone marrow, and it has been demonstrated that the cytogenetic alterations and mutations differ between different tumor sites demonstrating spatial genetic heterogeneity [18,19,20,38,39,40]. Therefore, single-site bone marrow aspirates do not address this spatial heterogeneity (Figure 2). The presence of a significant degree of subclonal heterogeneity, including site-specific subclones, adds another layer of genomic complexity to the characteristics of MM [20,23,39]. This subclonal diversity, with the presence of subclones harboring different genetic alterations conferring to therapy resistance, represents an important therapeutic challenge. The spatial genetic heterogeneity in the BM of MM patients and the dynamic evolution of the mutation profile throughout disease progression necessitate a flexible tool for characterization and follow-up. Because of the limitation to one single bone marrow site and the invasive nature of bone marrow aspirates, blood-based monitoring or liquid biopsies are a better tool to track the genetic changes over time. Also, they allow for non-painful, non-invasive, and frequently repeatable sample collection. An increasing number of studies support the applicability of circulating biomarkers for non-invasive genetic characterization in MM. Among the most studied biomarkers for this purpose are cfDNA and CTCs. These studies not only showed that these biomarkers can reliably reflect the tumor BM mutation profile but also highlighted the fact that circulating biomarkers can reveal genetic alterations not detected in matched BM samples [38,41,42,43,44,45,46,47,48,49]. In the following paragraphs, we will discuss these biomarkers and the evidence supporting their applicability in more detail.

## 4. Circulating Tumor Cells (CTCs) 

### 4.1. Characteristics 

One of the hallmarks characterizing malignant tumors is their ability to spread beyond the primary tumor site and cause disease dissemination at secondary locations. In MM, CTCs may play a central role in this process [50,51,52]. With average levels between 1 and 3.5 CTCs/µL, CTCs are detectable in virtually every MM patient with active disease when using sensitive techniques such as next-generation flow cytometry [48,50,51,53]. MM CTCs include PCs exhibiting phenotypically more immature characteristics with a decreased expression of integrins (CD11a, CD11c, CD29, CD49d, and CD49e) and adhesion molecules CD56 and CD117 [50,51]. This phenotype may result in a decreased capability to interact with the BM microenvironment, hence favoring escape into the bloodstream. Moreover, CTCs are associated with a quiescent state and an increased clonogenic potential, favoring the formation of secondary tumor lesions when re-entering the BM, and they might act as MM stem cells [50,51,54]. A recent study showed the overexpression of the stem cell marker CD44 and a decreased expression of genes related to proliferation (e.g., *CDC6*) in CTCs [55]. Interestingly, CTCs showed the overexpression of genes related to hypoxia (e.g., *DDIT4*) and epithelial–mesenchymal transition-related processes (e.g., *EMP3*), which are both hypothesized to contribute to the egression of CTCs out of their original BM niche and disease spreading [55,56].

### 4.2. Mutation Profiling in MM Using CTCs

The level of detectable CTCs is prognostically important, and high numbers confer to a faster disease progression and inferior survival rates in patients with MGUS, SMM, and MM [51,57,58,59,60]. Only a handful of studies so far investigated the applicability of CTCs for mutation profiling in MM (Table 1) [41,42,46,47,48]. Lohr et al. (2016) performed a single-cell mutational analysis on BM MM cells and CTCs isolated in 9 MM-patients [47]. They reported a 100% concordance between mutations found in BM MM cells and matched CTCs. Of note, the proportion of BM MM cells and CTCs harboring a mutation showed considerable intra- and interpatient variation and two mutations in *BRAF* and *NRAS* were only found in CTCs and not in matched BM MM cells. This study is, however, limited by the small number of patients (*n* = 9) and limited number of genes that were investigated and a moderate coverage depth. Mishima et al. (2017) used whole-exome sequencing (WES) to analyze the mutation profile in eight patients with paired BM and CTC DNA samples [46]. Although limited by the small sample size, an excellent concordance was observed, with 90% of CTC mutations (*n* = 572) detectable in BM and, inversely, 93% of BM mutations (*n* = 658) also detectable in CTCs. In addition, analysis of the cancer cell fraction (CCF) showed a near-100% concordance between clonal mutations detected in BM and CTCs, whereas 16% of the somatic variants were subclonal and only detectable in either BM or CTCs. In contrast to these results based on a limited number of matched CTC/BM MM cells samples used for sequencing, recent studies have expanded genetic analysis to larger patient cohorts [42,48]. Garcés et al. (2020) performed WES on 18 matched CTC/BM MM cell samples and reported an 82–86% concordance in the mutation profile. Interestingly, this analysis also included eight extramedullary (EM) plasmacytoma samples that were compared with matched CTC/BM MM cells. Here, an 87% concordance was observed between the mutation profiles in EM tumor cells and CTCs [48]. The ability of CTCs to permit the detection of mutations originating from the EM tumor compartment, which is often difficult to reach to conduct a biopsy, is an example of their added value. Moreover, CTCs might be responsible for the development of EM disease, given the high degree of mutational concordance between EM PCs and CTCs (e.g., shared mutations in *ZNF717*, *MTOR*, and *KLHL6*) [48,61]. It is, however, clear from the results discussed above that some degree of discordance between the mutation profiles in BM-DNA and CTCs is nearly always present. It is particularly interesting to determine whether mutations detected in BM but absent in matched CTCs could be detected in other sample types, such as cfDNA. This has been addressed in a very limited number of sequencing studies so far [41,42]. Of the clonal somatic variants present in the BM, Manier et al. (2018) detected 99% in cfDNA or CTC DNA in four patients with matched samples. However, several patients had somatic variants that were only detected in either cfDNA or CTCs [41]. It has recently been shown by Heestermans et al. (2022) that cfDNA allowed for the detection of 6/12 (50%) of BM+ somatic variants that were not detected in matched CTC samples [42]. These results highlight the fact that each biomarker can reveal unique variants that are not necessarily detectable in other matched circulating biomarkers, adding additional complexity to the mutational landscape in MM.

### 4.3. Biological Challenges in the Isolation of CTCs

To select and optimize a biomarker, technological challenges for isolation and use must be examined. The low absolute CTC counts in the blood of MM patients make their isolation and genetic interrogation challenging. In some of the studies cited above, CTCs were isolated from peripheral blood using an immunomagnetic enrichment method targeting the CD138 antigen [41,42,47]. Lohr et al. (2016) also targeted the CD45 antigen in their single-cell study [47]. However, the surface expression of CD138 in CTCs is lower compared to the clonal PCs in BM, which could hamper the efficiency of enrichment methods targeting this specific marker [50,62]. Currently, the CellSearch^®^ system is the only FDA-approved method for the capture and detection of CTCs that is intended for application in solid tumors [63]. Foulk et al. (2017) successfully applied this platform to isolate MM CTCs (defined as CD138+, CD38+, DAPI+, CD45−, and CD19−) and used them for downstream genetic analysis. Although targeting more surface markers than CD138 alone, the average recovery of spiked HMCL H929 cells was still limited to 61% [64].

Hence, the heterogeneous expression of CD138, CD38, and CD45 on malignant PCs represents a limitation for the use of enrichment methods to capture CTCs [65]. Other studies have used (next-generation) multiparametric flow cytometry (NGF) to sort CTCs based on aberrant patient-specific immunophenotypic features [46,48]. NGF could be of particular interest to detect MRD in the blood of MM patients by targeting CTCs, although current MRD assessment strategies still rely on BM evaluation [66]. The application of NGF requires prompt sample processing (as is the case for enrichment methods) and a high degree of local expertise to produce reliable results. Moreover, as the phenotype of malignant PCs can change over time after therapy exposure, this could make the detection of CTCs with NGF in a relapse setting more difficult [67]. Given the challenges associated with the isolation of CTCs, other circulating biomarkers like cfDNA that require less pre-analytical sample processing might be an interesting alternative for non-invasive genetic characterization in MM.

## 5. Cell-Free DNA (cfDNA)

### 5.1. Characteristics, Isolation Methods, and Associated Challenges

The presence of circulating nucleic acids in the blood of healthy individuals was first described in 1948 by Mandel and Métais [68]. In 1989, Stroun et al. discovered the presence of cfDNA originating from cancer cells in the blood of these patients, known as circulating tumor DNA (ctDNA) [69]. The discovery of cfDNA of fetal origin in the blood of pregnant women has led to the development of the NIPT test, which is now routinely performed in many countries for the prenatal detection of fetal trisomies [70]. Over the last decades, cfDNA has been increasingly studied as a non-invasive biomarker for genetic characterization and disease follow-up in MM and other cancers. The exact origin of cfDNA and ctDNA is still under debate, although it is generally accepted that cellular breakdown mechanisms such as necrosis and apoptosis play an important role in their release [71,72]. The typical peak in size distribution around 166 bp of double-stranded cfDNA fragments corresponds to the length of a nucleosome in which the DNA is cleaved after apoptosis, which supports this hypothesis. However, ctDNA fragments are typically shorter (<150 bp) [72]. Furthermore, it has also been suggested that the active release of cfDNA and ctDNA, e.g., associated with extracellular vesicles, plays an important role [72,73]. A plethora of commercially available kits for cfDNA extraction are currently available. Most isolation principles are based on binding the cfDNA to magnetic beads (e.g., Maxwell RSC kit^®^ (Promega)) or to columns coated with silica gel membranes (e.g., QIAamp circulating nucleic acid kit^®^ (Qiagen)) [74]. 

Because various (semi-)automated extraction devices are already available on the market, cfDNA has a high application potential in a clinical laboratory setting. However, a lack of standardization regarding the specimen collection and cfDNA extraction methods currently exists. This represents a major challenge in widespread clinical use, although recent harmonization efforts have been undertaken to provide guidelines for pre-analytical handling of cfDNA [75]. The choice of using serum or plasma for cfDNA extraction has an important impact on the results [76,77]. A higher degree of contamination with high-molecular-weight gDNA fragments is observed in serum compared to plasma [76,77]. This is most likely due to the release of cellular DNA from lysing WBC during the clotting process in serum specimens. The contamination with gDNA subsequently decreases the ctDNA detection rate and mutant allele fraction, hampering the clinical application of ctDNA [76,77]. Hence, plasma should be chosen over serum as the specimen type for cfDNA extraction. When considering the sample storage conditions prior to centrifugation and cfDNA extraction, plasma cfDNA levels remain stable for 24 h when whole blood is stored at 4 °C in K_2_-EDTA tubes [76]. In contrast, Cell-Free DNA BCT^®^ provides a cfDNA stability within the unprocessed specimen up to several days, even when stored at room temperature. Thus, using cold-stored EDTA tubes for cfDNA extraction is a valid and cost-efficient strategy only when immediate sample processing is possible (within 24 h after collection). If sample processing is delayed, then the use of cfDNA stabilizing tubes should be favored. 

The influence of the highly fragmented nature of cfDNA on the accuracy of commonly used quantification methods represents another challenge. Several studies have demonstrated that DNA fragmentation levels do not affect spectrophotometric quantification, whereas the performance of both fluorometric and qPCR-based quantification methods are considerably impaired in small DNA fragments with a size of around 150 bp [78,79]. Another limiting feature of ctDNA is its low abundancy. In most healthy individuals, cfDNA concentrations in the blood range between 0 and 100 ng/mL, and, in cancer patients, this generally increases to concentrations ranging between 0 and 1000 ng/mL [80]. The ctDNA fraction only represents a part of the total cfDNA. Lower ctDNA levels are measured in early tumor stages, and detectability may vary between patients and depend on the tumor type [81]. Hence, the isolation, quantification, and analysis of ctDNA is indeed challenging. However, a lot of progress has been made in the past decade in NGS technology, improving the sensitivity of ctDNA detection and downstream analysis. 

### 5.2. Mutation Profiling in MM Using cfDNA 

The use of cfDNA for mutation profiling in MM has been investigated by several research groups in the past few years [38,40,41,42,43,44,45,49,82,83,84,85]. Most studies so far used a targeted gene-sequencing strategy [38,42,43,44,49,83], while ddPCR [82,84] and WES [40,41,45] have also been used to a lesser extent. The observed proportion of variants found in BM-DNA that are also detectable in matched cfDNA samples varies between ±30% and 100%. However, most of the studies observed a > 80% concordance between BM-DNA and cfDNA [41,42,43,45,82,84]. This indicates that the BM-DNA mutation profile can be accurately characterized using cfDNA as a proxy. BM-DNA mutations not detected in cfDNA often have a low variant allele frequency (VAF) [42,43,44,49]. Hence, the detection of variants with a low tumor fraction in cfDNA is more challenging, as the ctDNA is diluted in a larger pool of cfDNA. On the other hand, nearly all studies report variants that are exclusively present in cfDNA and not detected in matched BM-DNA [38,40,41,42,43,45,49,82,83]. The most likely explanation for this is the spatial genetic heterogeneity in MM [18,19,20,38,39,40]. The ability to capture variants from distant tumor sites not reached by a single-site bone marrow aspirate represents a powerful advantage of using cfDNA. As EM disease is often located at body sites that are not easily accessible for a conventional biopsy, this advantage especially holds true for MM patients with EM disease [40]. Mithraprabhu et al. demonstrated major differences in the mutation profiles of BM and EM MM tumor sites and matched cfDNA in a patient with EM disease. Although a low-coverage WES approach was used, cfDNA permitted the detection of a major amount of the variants detected in BM/EM tissue, while also revealing unique variants detected in neither of the tumor tissues [40]. Taken together, the combined genetic analysis of cfDNA and primary tumor gDNA represents a valid strategy for comprehensive mutation profiling in MM patients. 

### 5.3. cfDNA-Based Monitoring of Treatment Response and Disease Progression

Several studies have reported that the longitudinal tracking of ctDNA can be informative in detecting MM disease progression or treatment response [38,40,41,43,45,83,84,85]. Changes in cfDNA tumor fraction and allelic frequency in specific mutations are often consistent with observable changes in standard measures of clinical progression (e.g., FLC ratio changes). Moreover, Mithraprabhu et al. observed that changes in ctDNA level can precede changes in serum FLC, thus permitting the earlier detection of disease progression [38]. This finding is supported by Rustad et al., who used ctDNA monitoring and showed that relapse could be detected three and nine months earlier in two MM patients, before serological changes became apparent [84]. In a patient with refractory EM disease, the levels of ctDNA continued to rise during progression, while FLC levels even paradoxically decreased [40]. Thus, in some cases, ctDNA-based monitoring outperforms conventional biomarkers to predict progression and trigger a change in therapy. Particularly in non-secretory MM, where no standard serological markers are available, serial analysis of cfDNA mutation status can be useful to evaluate response to therapy. Importantly, serial analysis of ctDNA can also reveal the appearance of new targetable mutations, providing an additional advantage.

## 6. Extracellular-Vesicle-Derived DNA (EV-DNA)

### 6.1. Characteristics of EVs and Applicability of EV-DNA for Mutation Profiling

The application of extracellular vesicles (EVs) as diagnostic and therapeutic tools and their role in cancer pathogenesis have gained substantial interest during the last decade. EVs are structures surrounded by a lipid bilayer. Depending on their size and cellular release mechanism, they classify as exosomes (30–200 nm) released by exocytosis of multivesicular bodies or microvesicles (100–1000 nm), which are released via budding and shedding of the plasma membrane. The largest EVs are apoptotic bodies (>1000 nm), which are released when cells undergo apoptosis [86]. Exosomes are thought to play an important role in tumorigenesis by acting as a signal carrier between tumor cells and the tumor microenvironment [87]. Currently, a multitude of EV isolation protocols exist that lack standardization. Most isolation methods involve differential centrifugation, density-gradient centrifugation, or ultracentrifugation. However, gel-permeation chromatography, membrane filtration, and recently developed microfluidic devices are also used, with varying success [88]. Interestingly, Thakur et al. demonstrated the presence of dsDNA in tumor-derived exosomes. The analysis of exoDNA originating from cancer cell lines permitted a detection of mutations with a 100% match to the mutation profile observed in the cell lines [89]. These results suggest that DNA derived from extracellular vesicles might be useful as a biomarker for mutation profiling. Indeed, in recent years, several studies were able to detect mutations in EV-DNA from patients with various cancer types [42,89,90,91,92,93,94,95,96]. Studies have successfully detected *KRAS* and *TP53* mutations in EV-DNA from patients with pancreatic cancer [90,91,94,95]. Similarly, Hur et al. and Wan et al. used exosome-derived DNA in NSCLC patients for *EGFR* genotyping [92,96]. Of note, in several studies EV-DNA even outperformed ctDNA in the detection of tumor-derived mutations [91,94,96]. As EV-DNA within the vesicles is protected by the lipid bilayer and EVs are secreted by metabolically active cells, EV-DNA might be a more stable and representative biomarker to study the tumor genetic profile. This is in contrast to cfDNA, which is unprotected and thus prone to rapid degradation, originating mainly from dying cells [97,98]. 

### 6.2. Challenges in the Isolation of EVs and EV-DNA

No standardized method for EV(-DNA) isolation is currently available, which hampers its clinical implementation as a diagnostic biomarker. The applicability of ultracentrifugation-based methods in clinical practice is limited, and the use of this technique as well as commercial isolation kits can result in the co-isolation of contaminating proteins and cfDNA [98,99]. This contamination together with the observation that only a part of the isolated EVs contain DNA can decrease the sensitivity of EV-DNA genotyping [99]. Thus, more research is needed to develop a method that can specifically capture EV-DNA-containing EVs. The inability of common EV isolation techniques to enrich tumor-derived EVs, which lowers the tumor purity of the sample, as well as the low EV-DNA yields that are typically obtained, further limit its application potential at the moment [42,100]. However, recent advances in microfluidics technology have led to the development of platforms such as ^new^ExoChip and OncoBean (DUO) that can specifically isolate tumor-derived EVs [101,102]. The exact EV-DNA packaging mechanism, as well as the size and location of the EV-DNA, is also still debated [98]. A recent study by Liu et al. found that EV-DNA can either be attached as relatively large DNA fragments to the surface of small EVs (<100 nm), as well as packed intravesically as smaller DNA fragments (200–1200 bp) in larger EVs (80–200 nm) [99]. This is contradictory to earlier reports showing that the DNA packed in larger EVs mainly consists of large DNA fragments with a size up to >2 Mbp [103]. Taken together, although the utility of EV-DNA as a cancer genetics biomarker has been shown, the limitations and challenges discussed above currently limit the widespread clinical implementation of EV-DNA as a diagnostic biomarker.

## 7. Towards Personalized Treatment in Multiple Myeloma Using Liquid Biopsies

The main clinical benefit of circulating biomarkers in MM is currently based on their prognostic potential, allowing the identification of mutations linked to inferior outcomes, as well as the detection of treatment failure and disease progression, as reviewed above. Their appropriate use may be helpful for adapting therapeutic strategies including the detection of targetable mutations. In MM, these mainly involve the *BRAF* V600E mutation, which is present in only a minority of patients [104]. Recently, studies have reported on selective inhibitors in *RAS*-mutated MM patients and the combined use of BRAF/MEK inhibitors in *BRAF*-mutated MM with promising results [105,106,107]. Although still far from clinical practice, circulating biomarkers could assess patient eligibility for these targeted treatments. More data relate to the way in which mutation profiling can be used to detect and predict therapy resistance [15,36,37]. Coffey et al. (2021) recently performed the targeted sequencing of cfDNA in 16 patients with RRMM and linked these results to high-throughput screening of drug compounds to predict response to these treatments. The authors of this study found that mutations in *DIS3*, *FGFR3*, *KMT2C*, *MAML2*, and *ZFHX4* were predictive of resistance to certain treatments, although no association was statistically significant after multiple-testing correction [108]. The monitoring of the genetic profile during treatment could be relevant in patients receiving CAR-T cell or bispecific T cell engager (TCE) therapies, as these are associated with high costs and potentially severe adverse effects. Of particular interest in this respect are two very recent studies, where resistance to anti-BCMA CART/TCE and anti-GPRC5D TCE therapy in MM patients could be linked to genetic (and/or epigenetic) alterations affecting the loci of these targets [109,110]. Similarly, using serial cfDNA analysis, Sworder et al. (2023) detected several mutations appearing de novo or clonally selected in large B cell lymphoma patients relapsing after anti-CD19 CAR-T cell therapy [111]. This study is a clear example of the usefulness of non-invasive molecular profiling and disease follow-up and how circulating biomarkers like cfDNA can be incorporated into a personalized treatment approach. As the number of MM patients receiving CAR-T/TCE cell therapies is increasing, the need for frequent sampling and molecular follow-up during treatment will become even more important, hence supporting the use of liquid biopsies to do so. 

## 8. Future Perspective

In the era of precision medicine, and with the emergence of novel treatment options in MM and other cancers, there is an increasing need for a reliable and practical tool for comprehensive tumor characterization and appropriate patient selection. The detection of (early) relapse and/or changes in the molecular properties of tumor cells might offer physicians useful information to select other (personalized) treatment options. Moreover, it might help us to understand the mechanisms underlying patients’ relapse when receiving novel immunotherapies. Liquid biopsies are a very advantageous strategy in MM, as they allow for more frequent sampling without the discomfort of repeated bone marrow aspirations. Because of the spatial genetic heterogeneity in the BM of MM patients, they are needed to obtain a more comprehensive overview of tumor-related genetic abnormalities. cfDNA appears to be the preferable biomarker for non-invasive mutation profiling in clinical practice, as we showed in a recent comparative study [42]. Its standardized and semi-automated isolation process, in particular, increases the clinical application potential of this biomarker. A very recently published paper by Kogure et al. (2024) constructed a prognostic index including tshe cfDNA mutation count and plasma cfDNA concentration, which could separate relapsed/refractory MM patients into different risk categories [17]. This gives an example of how prognostication based on cfDNA-based mutation profiling could be implemented into clinical practice in the near future. Beyond genetic alterations, MM is also characterized by distinct changes at the epigenetic level, which can have an impact on both prognosis and therapy response [112,113,114,115]. A spatial epigenetic heterogeneity in the bone marrow of MM patients has already been reported in the literature, although it is less well studied than the spatial heterogeneity at the genetic level [116]. Hence, liquid biopsies might again serve as a valuable alternative for BM aspirates to reliably and non-invasively characterize these epigenetic alterations. This has so far been investigated to a very limited extent in MM, and future (comparative) studies are needed to select the optimal (circulating) biomarker for this purpose [117,118]. The application of liquid biopsies for combined (epi)genetic characterization might be particularly relevant in the context of treatment with CAR-T cells or TCE, as previously referred to. 

## 9. Conclusions

In summary, in this review, we have discussed the characteristics and the applicability of blood-derived liquid biopsies for non-invasive molecular profiling in MM patients. We have addressed the limitations and challenges associated with their use and highlighted their potential to assess prognosis and assist in treatment decision making for MM patients.

## Figures and Tables

**Figure 1 ijms-25-05208-f001:**
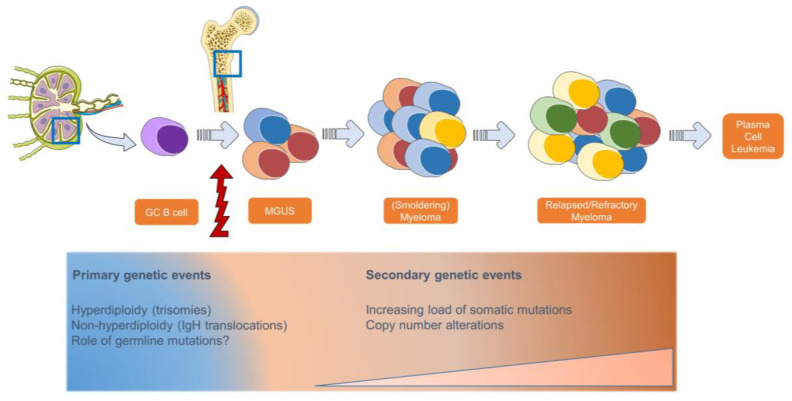
Evolving genetic profile in multiple myeloma throughout disease progression.

**Figure 2 ijms-25-05208-f002:**
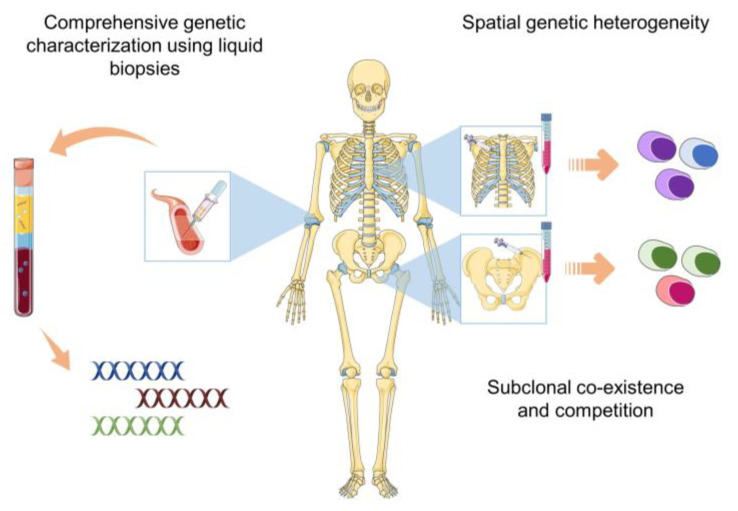
Liquid biopsies are non-invasive, practical tools to address the multi-level genetic heterogeneity in multiple myeloma.

**Table 1 ijms-25-05208-t001:** Overview of studies reporting mutation profiling data by analyzing CTC DNA samples. WGA: whole-genome amplification; ULP-WGS: ultra-low-pass whole-genome sequencing.

Study	Number of Patients	Methodology	Findings	Limitations
Lohr et al., 2016 [47]	9 patients with matched BM MM cells and CTCs	- Single-cell mutation analysis - Multiplex PCR—Ion Torrent PGM^TM^	- 12 mutations detected - All mutations found in BM MM cells also detected in matched CTCs - CTCs showed additional mutations not detected in BM MM cells	- Single-cell analysis less suited for routine laboratory practice - Moderate coverage depth - Low sample number - Low number of interrogated genes
Mishima et al., 2017 [46]	29 patients, including 8 patients with matched BM, CTC, and germline DNA samples	- WES on 13 CTC samples, including paired samples - Targeted gene sequencing on an additional 16 CTC samples	- 93% concordance between BM and CTC mutations - 16% of somatic mutations are subclonal and un-shared between BM PCs and CTCs	- Potential bias associated with use of WGA - Low number of matched BM/CTC samples
Manier et al., 2018 [41]	107 cfDNA and 56 CTC samples, including 4 patients with matched BM-cfDNA-CTC samples	- WES on matched samples - ULP-WGS for tumor fraction estimation	- In cases with matched BM-cfDNA-CTC, 99% of clonal mutations in BM also detected in cfDNA or CTCs - 6% of mutations detected in cfDNA and/or CTCs only	- Low number of matched BM/CTC/cfDNA samples - Limited applicability of WES
Garcés et al., 2020 [48]	53 patients with matched BM MM cells and CTCs (8 with EM disease sample), with mutation profiling in 18 patients	- WES preceded by WGA in 8 patients - WES preceded by molecular barcoding in 10 patients	- 82–86% concordance between BM and CTC mutations - 87% concordance between EM tumor cells and CTC mutations	- Limited applicability of WES due to costs and required expertise
Heestermans et al., 2022 [42]	30 MM patients, including 29 patients with matched BM DNA, cfDNA, EV-DNA, and CTC DNA	- Targeted gene sequencing with a 165-gene panel	- 83% concordance between BM and CTC mutations - 10% of mutations detected in CTCs but not in BM	- Limited number of interrogated genes - No estimation of tumor fraction performed because no WGS/WGA data available

## Data Availability

Data sharing is not applicable to this article.

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
