# Peer review of "Liquid Biopsies as Non-Invasive Tools for Mutation Profiling in Multiple Myeloma: Application Potential, Challenges, and Opportunities"

_ijms, 2024, doi:10.3390/ijms25105208_

Round 1

Reviewer 1 Report

Comments and Suggestions for Authors

The article "Liquid biopsies as non-invasive tools for mutation profiling in multiple myeloma: application potential, challenges and opportunities " thoroughly reviewed the current knowledge about the genetic and molecular features of MM which can be more effectively detected by the various modalities using circulating tumor cells (CTCs) and circulating cell free DNA (cfDNA) rather than conventional analysis with "one-site" biopsied specimen. The article nicely reviewed the advantage and the disadvantage of the analysis using CTCs and cfDNA, addressing its potential for the more accurate and precise understanding disease pathophysiology. This may make the article more informative to readers.

One point I would address is, although as its nature as the review article, the article is somewhat descriptive. Please add the examples of important genes detected in CTCs of cfDNA in several particularly important settings listed as below.

1.  Page 4. Authors described that overexpression of the stem cell marker CD44 and decreased expression of genes related to proliferation in CTCs [47], and that CTCs also showed overexpression of genes related to hypoxia and epithelial-mesenchymal transition-related processes, which are both hypothesized to contribute to the egression of CTCs out of their original BM niche and disease spreading [47. 48]. Please raise some critically important  genes as examples, in addition to CD44.

2. Page 5. Authors suggested that CTCs might be responsible for the development of EM disease, given the high degree of mutational concordance between EM PCs and CTCs. which may be different from BM PCs. Please raise some likely important genes, as examples.

3. Page 8. Authors raised the concern about major differences in the mutation profiles of BM and EM MM tumor sites and matched cfDNA in a patient with EM disease. Please raise several likely important genes, as example.

Author Response

Response to Reviewer 1 Comments

Response to the reviewer (summary):

The authors thank the reviewer for the critical and constrictive review and all the interesting questions and  suggestions. We have done our best effort to adapt the manuscript accordingly. Below, the specific answers to the different comments can be found.

 Point-by-point response to Comments and Suggestions for Authors:

  1. Page 4. Authors described that overexpression of the stem cell marker CD44 and decreased expression of genes related to proliferation in CTCs [47], and that CTCs also showed overexpression of genes related to hypoxia and epithelial-mesenchymal transition-related processes, which are both hypothesized to contribute to the egression of CTCs out of their original BM niche and disease spreading [47. 48]. Please raise some critically important genes as examples, in addition to CD44.

Response to the reviewer:

We agree that adding more information about specific genes differentially expressed in CTCs would make this paragraph more informative. Examples of specific genes were added to line 140 and 142 of the revised version of the manuscript.  

  1. Page 5. Authors suggested that CTCs might be responsible for the development of EM disease, given the high degree of mutational concordance between EM PCs and CTCs. which may be different from BM PCs. Please raise some likely important genes, as examples.

Response to the reviewer:

Although the number of studies investigating the concordance between the mutation profiles of CTCs and EM PCs is limited, both cited studies reported shared mutations in ZNF717. This protein plays a role in cell proliferation, differentiation and apoptosis. Other genes showing shared mutations between CTCs and EM PCs included MTOR and KLHL6, both of which are associated with cancer and/or MM in particular. The above-mentioned examples of genes with shared mutations were added to line 173-174 of the revised version of the manuscript.

  1. Page 8. Authors raised the concern about major differences in the mutation profiles of BM and EM MM tumor sites and matched cfDNA in a patient with EM disease. Please raise several likely important genes, as example.

Response to the reviewer:

Unfortunately, no details about the specific genes differentially mutated between the tumor sites (BM versus EM tumor site versus cfDNA) can be retrieved from the source publication (Mithraprabhu et al., Int. J. Mol. Sci., 2018). It only states the number of SNVs retrieved across the different locations at initial evaluation (935 SNVs), of which 42% were found in only one tumor site. This suggests that indeed important differences can exist between the mutation profile of BM versus EM tumor site versus cfDNA in blood, and further studies are needed to investigate this.

Reviewer 2 Report

Comments and Suggestions for Authors

I found this as a solid work and its quality is good, however, the author still needs to solve the following problems.

1-      The introduction section should be updated by data more related and updated data from

Scopus.

2-      The structure of the manuscript is not clear. Sections are not well-defined.

3-      Utilization of next-generation sequencing in multiple myloma should be added in a separate paragraph.

4-      In the 7th part of the paper (Towards Personalized Treatment in Multiple Myeloma Using Liquid Biopsies), the importance and unique value of liquid biopsy-based treatments for MM should be strengthened by providing a table.

5-      A paragraph on therapeutic monitoring should be added.

6-      In addition, please make new paragraph about the perspective.

7-      The conclusion section should be concise.

8-      Do risk factors and pathogenic mechanisms of MM vary between age groups, ethnicities or disease states?

9-      Provide a table of abbreviations with their expansions.

10-  Numerous researches have examined the role of liquid biopsies for mutation profiling in multiple myloma. What benefits does this manuscript have over other research?

11-  Advantages of liquid biopsies over other laboratory examination should be discussed.

12-  I think the author could elaborate the genetic variation section into various subsections like different translocations, prognosis and its therapeutic implications, deletions, etc.

13-  Add clinical relevance of liquid biopsies.

14-  Detection techniques for circulating tumor cells should be added.

15-  The biggest weakness of the manuscript is the limited clinical data and which should be included in the medical-related manuscript.

Comments on the Quality of English Language

Minor editing of syntax and grammar is advised.

Author Response

Response to Reviewer 2 Comments

Response to the reviewer (summary):

The authors thank the reviewer for the critical and constrictive review and all the interesting questions and suggestions. We have done our best effort to adapt the manuscript accordingly. Below, the specific answers to the different comments can be found.

Point-by-point response to Comments and Suggestions for Authors:

  1. The introduction section should be updated by data more related and updated data from Scopus.

Response to the reviewer:

The authors agree with the reviewer that the introduction section could benefit from some more recent and up-to-date references to papers supporting the text. Therefore, the following recent references were added to the introduction section, all of which are indexed in Scopus:

  • van de Donk, N.; Pawlyn, C.; Yong, K.L. Multiple myeloma. Lancet 2021, 397, 410-427, doi:10.1016/s0140-6736(21)00135-5. (line 30)
  • Rajkumar, S.V. Multiple myeloma: 2022 update on diagnosis, risk stratification, and management. Am J Hematol 2022, 97, 1086-1107, doi:10.1002/ajh.26590. (line 40)
  • Kogure, Y.; Handa, H.; Ito, Y.; Ri, M.; Horigome, Y.; Iino, M.; Harazaki, Y.; Kobayashi, T.; Abe, M.; Ishida, T.; et al. ctDNA improves prognostic prediction in relapsed/refractory MM receiving ixazomib, lenalidomide, and dexamethasone. Blood 2024, doi:10.1182/blood.2023022540. (line 43)
  1. The structure of the manuscript is not clear. Sections are not well-defined.

Response to the reviewer:

The authors agree with the reviewer that the titles of some of the sections of the paper could be improved or clarified to make the manuscript more structured. Therefore, we changed the title of section 2 (line 56-57) and added additional subsections 6.1 and 6.2 with explanatory title (line 324 and 351) to the revised version of the manuscript.

  1. Utilization of next-generation sequencing in multiple myeloma should be added in a separate paragraph.

Response to the reviewer:

The authors agree with the reviewer that the addition of a small piece of text putting the use of NGS in MM clinical practice into perspective would improve the manuscript. A paragraph about the current use for NGS in clinical practice in MM was added to the revised version of the manuscript, line 79-85, together with supporting literature references. The general topic of the application of NGS in MM has been extensively reviewed before, and is cited in the revised version of the manuscript. We think that further elaboration about this topic would be out of the scope of this paper.

  1. In the 7th part of the paper (Towards Personalized Treatment in Multiple Myeloma Using Liquid Biopsies), the importance and unique value of liquid biopsy-based treatments for MM should be strengthened by providing a table.

Response to the reviewer:

As far as we are aware of, liquid biopsy-are not yet used in clinical practice to direct the course of treatment in MM. The cited studies in this section relate to the way mutation profiling can be informative to assess patient eligibility or detect mechanisms of relapse in patients receiving treatment. A major part of the cited studies in this section report on results obtained by using bone marrow aspirates. The studies by Coffey et al. (2021) and Sworder et al. (2023) report on results obtained with cfDNA. Both of these studies are discussed in more detail in this paragraph, and give an impression on how liquid biopsies might be contributive to personalized medicine (line 384-389; line 394-399). Hence, we think that adding an additional table to include the results of these studies would not provide additional information to the reader. We would therefore prefer not to include an additional table in this part of the paper.

  1. A paragraph on therapeutic monitoring should be added.

Response to the reviewer:

Indeed, the applicability of liquid biopsies for therapeutic monitoring and detection of early relapse is a key advantage of using this strategy. This item has been addressed in the manuscript under section 5.3 “cfDNA-Based Monitoring of Treatment Response and Disease Progression”, line 306-322 of the revised manuscript.

  1. In addition, please make new paragraph about the perspective.

Response to the reviewer:

The authors thank the reviewer for this comment, which indeed improves the manuscript. A separate section “Future Perspective” was added to the revised version of the manuscript, line 403-431. The first part of this paragraph consists of the author’s view on the topic, and continues with directives for future research, focusing on the application of liquid biopsies for prognostication in clinical practice as well as the characterization of the epigenetic changes in MM.

  1. The conclusion section should be concise.

Response to the reviewer:

A revised, more concise version of the conclusion was added as section 9, line 432-437 to the revised version of the manuscript.

  1. Do risk factors and pathogenic mechanisms of MM vary between age groups, ethnicities or disease states?

Response to the reviewer:

It is well known that the incidence/risk of MM is higher among men and non-Hispanic black people, which has been attributed to a higher incidence of MGUS in these populations. The underlying pathogenic mechanisms are not yet fully understood, but differences in inherited susceptibility loci are thought to play a role in this (reviewed by Smith et al., Blood Cancer Journal, 2018). A sentence describing the differences in MM incidence based on gender and ethnicity has been added to the revised version of the manuscript, line 34-36.

  1. Provide a table of abbreviations with their expansions.

Response to the reviewer:

In agreement with the instructions for authors from IJMS, all abbreviations that were used are added in parentheses after the written-out form when they first appear in the text. Hence, we think that adding a separate table with abbreviations would not give additional information to the reader.

  1. Numerous researches have examined the role of liquid biopsies for mutation profiling in multiple myeloma. What benefits does this manuscript have over other research?

Response to the reviewer:

In this paper, we critically review research findings about the clinical validation and applicability of circulating biomarkers for mutation profiling in MM. This review is based on the most important literature that has been published over this subject during the past decade. The applicability of EV-DNA for mutation profiling in MM and other cancers is extensively discussed, which is often omitted from other review papers (who focus merely on cfDNA and CTCs). Moreover, this paper discusses some of the most recently published results in the fields (papers published up to March 2024) and thus provides an up-to-date overview of the relevant literature. We believe that this is of added value to the reader.

  1. Advantages of liquid biopsies over other laboratory examination should be discussed.

Response to the reviewer:

The added value of liquid-biopsy based mutation profiling compared to standard serological examinations (e.g. follow-up of the M-protein and/or FLC in blood) has already been addressed in the manuscript (line 311-322).

  1. I think the author could elaborate the genetic variation section into various subsections like different translocations, prognosis and its therapeutic implications, deletions, etc.

Response to the reviewer:

The authors agree with the reviewer that some additional information about the cytogenetic alterations (including the translocations, deletions, etc.) would be interesting for the reader. The most important are the adverse risk cytogenetics. Hence, this information was added to the revised version of the manuscript (line 65-68). However, we think that a further detailed discussion of each individual high risk cytogenetic alteration would deviate too much from the focus of this review paper on the characterization of the mutation profile in MM.

  1. Add clinical relevance of liquid biopsies.

Response to the reviewer:

The rationale for the use of liquid biopsies and there added clinical value and relevance over conventional bone marrow aspirates has been discussed extensively in section 3 of the manuscript (line 100-122). A detailed discussion about the clinical relevance of liquid biopsies with examples is integrated in the different sections of the manuscript. In line 146-148, the prognostic value of CTC levels in MM has been addressed. The specific advantages of liquid biopsies to permit detection of variants not found at single bone marrow sites or to permit mutation profiling in patients with extramedullary disease (which is difficult to reach for a biopsy) is addressed into more detail in line 293-299. Moreover, section 5.3 “cfDNA-Based Monitoring of Treatment Response and Disease Progression “, discusses the ability of cfDNA to monitor treatment response and detect early disease progression, which are both examples of clinical relevance of liquid biopsies (line 306-322). Finally, line 394-399 and line 417-421 discuss the real-world application of liquid biopsies for prognostication and personalized treatment.

  1. Detection techniques for circulating tumor cells should be added.

Response to the reviewer:

We would like to refer to section 4.3 “Biological Challenges in the Isolation of CTCs“ where different capturing/detection techniques for CTCs are extensively discussed. For the current manuscript, we focused on the detection methods that are relevant for the study of CTCs in MM and have been used for this purpose in previous studies. This includes the immunomagnetic separation of CTCs (line 188-200) and detection via next generation multiparametric flow cytometry (line 216-224).

  1. The biggest weakness of the manuscript is the limited clinical data and which should be included in the medical-related manuscript.

Response to the reviewer:

All relevant studies related to the clinical, prognostic and therapeutical relevance of liquid biopsies in MM have been addressed in the current manuscript. The clinical relevance and application of liquid biopsies has been addressed on multiple locations in the manuscript (line 146-148; line 293-299; line 306-322; line 417-421). For a more detailed description of the paragraphs addressing the clinical relevance, we would like to refer to the answer to comment 13 of the reviewer. Moreover, the clinical relevance of mutation profiling in MM, including mutations that are associated with therapy resistance and prognostic impact, has been discussed in detail in section 2 of the manuscript (line 85-97).

Round 2

Reviewer 2 Report

Comments and Suggestions for Authors

I would like to thank the authors for addressing my initial comments. The authors have very effectively addressed all comments. Following the revision to the article, I feel that this manuscript is now acceptable for publication.